



# The Strengthening Relationship between Eurasian Snow Cover and December Haze Days in Central North China after the Mid-1990s

Zhicong Yin [12] and Huijun Wang [12]

[1]Key Laboratory of Meteorological Disaster, Ministry of Education / Joint International Research Laboratory of Climate and Environment Change (ILCEC) / Collaborative Innovation Center on Forecast and Evaluation of Meteorological Disasters (CIC-FEMD), Nanjing University of Information Science & Technology, Nanjing 210044, China

[2]Nansen-Zhu International Research Centre, Institute of Atmospheric Physics, Chinese Academy of Sciences, Beijing, China

*Correspondence to*: Zhicong Yin (yinzhc@163.com)

**Abstract.** The haze pollution in December has become increasingly serious over recent decades and imposes damage on society, ecosystems, and human health. In addition to anthropogenic emissions, climate change and variability were conducive to haze in China. In this study, the relationship between the snow cover over East Europe and West Siberia ($SC_{ES}$) and the number of haze days in December in central North China was analyzed. This relationship significantly strengthened after the mid-1990s, which is attributed to the effective connections between the $SC_{ES}$ and the Eurasian atmospheric circulations. During 1998–2016, the $SC_{ES}$ significantly influenced the soil moisture and land surface radiation, and then, the combined underlying drivers of enhanced soil moisture and radiative cooling moved the East Asia jet stream northward and induced anomalous, anti-cyclonic circulation over central North China. Modulated by such atmospheric circulations, the local lower boundary layer, the decreased surface wind and the more humid air were conducive to the worsening dispersion conditions and frequent haze occurrences. In contrast, from 1979 to 1997, the linkage between the $SC_{ES}$ and soil moisture was negligible. Furthermore, the correlated radiative cooling was distributed narrowly and far from the key area of snow cover. The associated atmospheric circulations with the $SC_{ES}$ were not significantly linked with the ventilation conditions over central North China. Consequently, the relationship between the $SC_{ES}$ and the number of hazy days in central North China was insignificant before the mid-1990s but has strengthened and has become significant since then.

**Keywords.** Haze, Aerosol, Snow cover, North China, Climate Variability





## 1. Introduction

In December 2016, central North China (CNC, located at 30–41 °N, 110–120 °E), where more than 300 million people live, experienced severe haze pollution (Yuan and Ma 2017). On 70% of the days in December 2016, the people who lived in CNC breathed polluted air, which influenced the health of everyone, especially children. Beyond anthropogenic emissions, the influences of atmospheric circulations on this severe haze event were significant (Yin and Wang 2017b). Many previous studies have documented that climate change and variability contributed to the severe winter haze pollution in China (Cai et al. 2017; Ding et al. 2014; Wang and Chen, 2016; Yang et al. 2016). The possible physical processes in the atmosphere that caused this event may have included, when the positive pattern of East Atlantic/West Russia and West Pacific (Yin et al. 2017a) occurred together or partly, the anomalous anti-cyclone over CNC and Japan Sea would be enhanced and then confined the vertical motion of atmospheric matters. Furthermore, the southerly anomalies that are characteristic of East Asian winter monsoons (Li et al. 2015; Yin et al. 2015) may have weakened the cold air and wind speed but enhanced the transportation of humid air flow. Thus, the vertical and horizontal dispersion capacities were both restricted, which resulted in haze pollution. In regard to external mechanisms, the investigated climatic factors include sea surface temperature (SST) over the North Atlantic (Xiao et al. 2015) and subtropical Western Pacific (Yin and Wang 2016, Gao and Chen 2017), Arctic sea ice (Wang and Chen 2015, 2016) and the topography of the Tibetan Plateau (Xu et al. 2016). In addition, the large-scale SST patterns, such as the El Nino-Southern Oscillation and the Pacific decadal oscillation, also showed close relationships with the haze pollution in the east of China (Gao et al. 2015).

Different from the declining trend of Arctic sea ice, Eurasian snow cover has been increasing over the last two decades (Cohen et al. 2012). The anomalous snow cover influenced the exchange of heat and moisture in atmosphere-land interactions, which were characterized by high albedo and water effects (Chen et al 2012). Starting in autumn, the snow cover over Eurasia began to accumulate gradually and was significantly correlated with the winter climate in the Northern Hemisphere (Foster et al. 1983; Zhang et al. 2007; Li and Wang 2014; Li et al 2017; Xu et al. 2017). In October, enhanced snow cover was associated with a negative Arctic Oscillation phase (Gong et al. 2007) via the stratosphere–troposphere coupled planetary wave activity (Cohen et al. 2007). The change in the October-November (ON) Eurasian snow cover was also considered as a primary factor for the recent recovery of the Siberian High intensity over the last few decades (Jeong et al. 2011). Furthermore, there was a significant negative correlation between the October snow cover located in eastern Siberia and in the area northeast of Lake Baikal and the following-winter air temperature over Northeast China (Li et al.



2017). A notable feature related with the impact of snow cover was the change in the relationship with the winter climate in the Northern Hemisphere after the mid-1990s. Both observational evidence and model simulations demonstrated a significant change in the relationship between the autumn Eurasian snow depth and the East Asian winter monsoon (Li and Wang 2014). Xu et al (2017) applied a 15-year sliding correlation to show the intensification in the connection between the October snow cover and the January "warm Arctic-cold Eurasia" pattern since the mid-1990s. Specifically investigating the impact of snow cover on December haze days over the CNC area ($DHD_{CNC}$), Yin and Wang (2017b) illustrated that $DHD_{CNC}$ significantly related with the ON snow cover over East Europe and West Siberia ($SC_{ES}$). Zou et al (2017) also pointed out that there was close relationship between Eurasia snow and haze in China basing on the observational and numerical analysis.

Thus, a question raised here was whether there was a significant change in the connection between $SC_{ES}$ and $DHD_{CNC}$. Motivated by many previous studies, we attempted to answer this question and explored the associated physical mechanisms. The investigation described in this paper will highlight the impact of $SC_{ES}$, recognize the changes in their relationships with other variables, and improve the seasonal prediction potential of the $DHD_{CNC}$.

The remainder of this paper is organized as follows. The data and methods are described in section 2. In section 3, we analyzed the strengthening relationship between $SC_{ES}$ and $DHD_{CNC}$, as well as the associated atmospheric circulations. Then, the possible physical mechanisms were studied in section 4. The main conclusions of this study and necessary discussion material are included in section 5.

## 2. Datasets and methods

The geopotential height at 500 hPa (Z500) and 200 hPa (Z200), the zonal wind at 200 hPa (U200), the wind at 850 hPa (UV850), the wind speed at the surface, the sea level pressure (SLP), the surface air temperature (SAT), the surface relative humidity, the vertical wind, the net longwave radiation and the net shortwave radiation data were downloaded from the National Center for Environmental Prediction and the National Center for Atmospheric Research. These 2.5 °×2.5 °reanalysis datasets were available for the period between 1948 and 2016 (Kalnay et al. 1996). In addition, the 1 °×1 °planetary boundary layer height (BLH) was derived from the ERA-Interim dataset (Dee et al. 2011). The monthly snow cover data were supported by the Rutgers University (Robinson et al. 1993). The sub-daily (i.e., four times per day) routine meteorological observations (i.e., relative humidity, visibility, wind speed, and weather phenomena) were collected by the National Meteorological Information Center, China Meteorological Administration. According to Yin et al. (2017a), the haze data





were calculated mainly based on the observed visibility and the relative humidity. Because the interval of the haze data was six hours, we defined a haze day as a day with haze occurring at any of the four times. $DHD_{CNC}$ was the mean number of

80    haze days over the CNC area.

### 3.    Strengthening relationship and associated atmospheric circulations

From 1979 to 1997, interannual variation was the main change mode of the $DHD_{CNC}$, and the linear sloped trend was not significant (Figure omitted). Thereafter, the decadal component of the $DHD_{CNC}$ became significant; that is, the haze days decreased from 1998 to 2010 but increased rapidly (Figure 1a), reaching more than 21 days in 2016. The minimum number

85    of $DHD_{CNC}$ was 10 days and occurred in 2010, while the maximum, 21 days, appeared in 2016 (Yin and Wang 2017b). As illustrated by Yin and Wang (2017b), the $DHD_{CNC}$ has a significantly close relationship with $SC_{ES}$ between 1979 and 2016 (Figure 1b); $SC_{ES}$ was defined as the area-averaged ON snow cover over East Europe and West Siberia (ES: 40–90 °E, 50–60 °N). This domain was consistent with the centers of the dominant varied mode calculated by Sun (2017). The positive correlation meant that if there was more $SC_{ES}$, then the haze pollution would be more severe over the CNC area. From the

90    perspective of temporal variation, the $SC_{ES}$ was more consistent with the $DHD_{CNC}$ after the mid-1990s. Similar to the $DHD_{CNC}$, the maximum and minimum values of the $SC_{ES}$ were also observed in 2016 and 2010, respectively. It appeared that the correlation before the mid-1990s was not significant. Chronologically, the $SC_{ES}$ decreased from 2000 to 2010, but it increased thereafter, which was similar to the $DHD_{CNC}$. Thus, the 21-yr running correlation coefficient (CC) between the $DHD_{CNC}$ and $SC_{ES}$ was calculated and plotted in Figure 1a. Obviously, the CC was strengthened and became significant after

95    the mid-1990s, exceeding the 99% confidence level. The CC between the $DHD_{CNC}$ and $SC_{ES}$ during the period between 1998–2016 (P2) was 0.62 after detrending, which was more significant than that during the period between 1979–1997 (P1), i.e., only 0.07. The ON Eurasian snow cover correlated with the $SC_{ES}$ was greater, and the CC was also larger during P2 (Figure 2), indicating that the snow cover varied more within the key areas and could influence the local and teleconnected climate more significantly. However, during P1, the CC over the east part of the ES area was insignificant. The interannual

100    variations in snow cover over the Tibet Plateau and Mongolian Plateau were evident both during P1 and P2. In contrast, the interannual variation in snow cover over East Europe and West Siberia was larger during P2 than during P1. Furthermore, during P2, the snow cover with larger interannual variation was distributed widely and zonally; in contrast, during P1, the significantly varied snow cover was meridionally instead of zonally and was only located to the north of the Black Sea; thus,



it could not have been teleconnected with the haze pollution in China. We speculated that the varied interannual variation of the $SC_{ES}$, which was also revealed by empirical orthogonal function analysis (Figure omitted), possibly influenced the strengthening relationship shown in Figure 1a. The impact of Arctic amplification on East Asian winter climate was significant (Wang and Liu 2016; Zhou 2017). Wang et al (2015) illustrated the decline in Arctic sea ice intensified the haze pollution in eastern China. Thus, we calculated the CC between the $SC_{ES}$ ($DHD_{CNC}$) and the Arctic sea ice during P1 and P2, respectively. The $SC_{ES}$ was insignificantly correlated with the September–November sea ice during P1 (Figure S1) but was significantly correlated with the ON sea ice over the Barents Sea (above 95% confidence level) during P2 (Figure S2). However, during P2, the CC between the ON sea ice over the Barents Sea and the $DHD_{CNC}$ was not significant, indicating that the Eurasian snow cover was relatively independent of the Arctic sea ice in terms of its impact on haze pollution over the CNC area.

To explore the reasons for the observed strengthening relationship, the associated atmospheric circulations with the $SC_{ES}$ during P2 are shown in Figures 3-5. In the upper troposphere, the induced centers of atmospheric activities appeared as a "+–+–" pattern, including the positive centers located in West Europe, North China and the Japan Sea and the negative centers over the north of the Caspian Sea and Aleutian Islands (Figure 3). This Rossby wave-like pattern also existed and propagated with observed wave activity flux in the mid-troposphere (Figure 4). The positive anomalies over North China and the Japan Sea were connected with the subtropical high in the Pacific, resulting in a strong pressure gradient in the south of the Aleutian low. The East Asia jet stream (EAJS), particularly its western end, was located more northward, meaning that the activities of the Rossby waves were also located more northward, and the cold air moving southward to the CNC area was weak (Chen and Wang 2015). The anomalous anti-cyclone over the CNC area could be observed in the upper (Figure 3), middle (Figure 4) and lower troposphere (Figure 5b). The sinking motion caused by these anti-cyclonic anomalies could lead to the shallower planetary boundary layer (Figure 5a) and the rather weak dispersion capacity of atmospheric particulates. In contrast, the associated vertical velocity at the surface was upward, indicating an ascending motion near the surface. The local rising-air, combined with the weak south wind, easily enabled aerosols to accumulate over the CNC area. Near the surface, the positive SLP anomalies were situated in the east of China and the western Pacific (Figure 5c). The stimulated southerlies overlapped with the mean flow of the East Asian winter monsoon to weaken the cold northerly winds. The SAT of Eurasia was warmer, and the surface wind speeds over the CNC area were significantly reduced; thus, the horizontal ventilation capacity of the atmosphere over the CNC area was weak, and it was difficult for the air pollutants to disperse.





Moreover, the enhanced water vapor transportation by the anomalous southerlies provided a beneficial environment for hygroscopic growth, which is an important process for the formation of severe haze pollution. In summary, during P2, the atmospheric circulations and local meteorological conditions, which were related with the $SC_{ES}$, effectively confined the vertical and horizontal dispersion of atmospheric particles.

For comparison, the associated atmospheric circulations during P1 are shown in Figures 6–8. In the mid and high troposphere, the zonal Rossby wave-like pattern, which existed during P2, could not be identified; rather, another pattern propagated meridionally from the Mediterranean Sea to the polar region and then through Northeast China and the Okhotsk Sea to the West Pacific (Figure 6–7). The wide and zonal cyclonic anomalies located over Northeast China and the Okhotsk Sea strengthened the EAJS and the meridional movement of cold air and resulted in the lower SAT in the east of China. The

associated anomalous circulations tended to lead local meteorological conditions (e.g., higher BLH and more obvious surface wind speed) to favor ventilation, which was consistent with the 21-yr running CC in Figure 1a (i.e., negative before the mid-1990s). However, this negative relationship was not significant because the correlated area of BLH and surface wind was too narrow; additionally, the surface vertical motion and relative humidity were not significantly correlated with the $SC_{ES}$ during P1.

**4.    Possible physical mechanisms**

In autumn, the snowfall began in the mid-high latitudes. Because the SAT was not persistently below freezing point, part of the snow melted, and the soil moisture increased. In addition to the snow melt, the accumulated snow cover also reduced the moisture that evaporated from the land surface. During P2, the $SC_{ES}$ was significantly positively correlated with soil moisture around the Caspian Sea, Balkhash Lake, and Ural Mountains (RM1: 50–80ºE, 40–60 ºN). In addition, when the

$SC_{ES}$ was greater, the soil was drier to the northeast of Lake Baikal (RM2: 100–130 ºE, 52.5–62.5 ºN). These two significant correlations persisted and were enhanced in December, i.e., the CC between the ON snow cover and the December soil moisture was larger than that between the ON snow cover and the ON soil moisture, both in the RM1 and RM2 areas. The area-averaged soil moisture in RM1 (RM2) was denoted as $SoilM_{RM1}$ ($SoilM_{RM2}$), and the SoilM index was defined as the difference between $SoilM_{RM1}$ and $SoilM_{RM2}$ (i.e., $SoilM = SoilM_{RM1}-SoilM_{RM2}$). During P2, the CC between the $DHD_{CNC}$

and the ON (December) SoilM index was 0.69 (0.69) after the removal of the linear trend, and it exceeded the 99% confidence level; however, these significant correlations did not exist during P1 (Table 1). We speculated that the ON snow





cover could impact the soil moisture in the RM1 and RM2 areas, which could last into December, and then influence the December haze pollution through atmospheric circulations. Thus, the associated atmospheric circulations in the mid-troposphere were calculated and shown in Figure 10. During P1, the impacts of the SoilM index on Z500 were not

significant in ON or December, but this was consistent with the weak relationships between the $SC_{ES}$ and $DHD_{CNC}$. In contrast, the significantly induced atmospheric circulations were distributed as a zonal Rossby wave-pattern during P2 (Figure 10 b, d), which is similar to the data shown in Figure 4. Particularly, the anomalous anti-cyclonic circulation over the CNC area was significant both in ON and December and was connected with the weak dispersion capacities of the atmospheric particles. The possible physical processes causing this could include the larger snow cover increasing the local

soil moisture by melting and impeding evaporation, and the wetter land surface may have persisted and been enhanced in December. The "west wet-east dry" pattern of soil moisture could influence the atmospheric circulations, which would benefit the occurrence of haze pollution as a result of poor dispersion conditions. During P1, both the CC between the $SC_{ES}$ and the SoilM index and the CC between the SoilM index and the $DHD_{CNC}$ were not significant, indicating that the snow cover in the study area did not impact the $DHD_{CNC}$ through effects on land surface moisture.

High albedo is another obvious characteristic of snow cover, which reflects more solar shortwave radiation and results in lower SAT. As a feedback, the outgoing longwave radiations emitted by the cooler land surface were weakened and had radiative cooling impacts on the atmosphere (Zhang et al. 2017). As shown in Figure 11, the correlated areas of radiation, including the location and shape, were apparently different during these two periods. During P1, the significant CCs between the $SC_{ES}$ and net shortwave radiation were distributed from the southwest (i.e., Pamir Mountains) to the northeast (i.e., Sayan

Mountains); this was denoted as RS1 (70–100 °E, 38–58 °N) and was mountainous. In contrast, the regions that had significant and negative CCs and net longwave radiation were smaller and over the Pamir Mountains (RL1: 67.5–90 °E, 36–45 °N). By contrast, the significant correlated regions with net longwave radiation (RL2) and net shortwave radiation (RS2) were the same and nearly overlapped with the ES area during P2, which was wider and had a zonal distribution. According to the above analysis, if there was more $SC_{ES}$, the absolute value of the net longwave radiation and net shortwave

radiation would both be smaller. To assess the combined effects of radiation, the $I_{LS}$ index was defined as the sum of the absolute value of the area-averaged net shortwave radiation ($|I_{RS}|$) and the absolute value of the area-averaged net longwave radiation ($|I_{RL}|$), i.e., $I_{LS1} = |I_{RL1}| + |I_{RS1}|$ and $I_{LS2} = |I_{RL2}| + |I_{RS2}|$. It was obvious that when there was more snow cover, the $I_{LS1}$ and $I_{LS2}$ should be smaller. After removing the linear trend, the CCs between $I_{LS}$ and $DHD_{CNC}$ were calculated and were 0.07 ($I_{LS1}$,





not significant) and –0.72 ($I_{LS2}$, above 99% confidence level). We speculated that the ON snow cover would influence the

December haze pollution by modulating the radiation during P2, but this process did not exist during P1. In fact, Cohen et al

(2007) noted that the diabatic cooling in late autumn, which was in accordance with the higher-than-normal snow cover,

locally induced higher SLP anomalies and colder SAT; then, significant influences on the tropospheric atmosphere were

observed the following winter through stratosphere-troposphere coupling. During P2, because of radiative cooling, the ON

SAT was lower over the ES area and was zonally spread to the Okhotsk Sea. Positive ON SLP anomalies were also

stimulated in the mid-high latitudes of Eurasia. The induced SLP and SAT anomalies were zonal and almost occupied the

mid-high latitudes of Eurasia. In contrast, the SLP and SAT anomalies during P1 were more meridional and smaller and were

more westward and located over Europe (Figure 12). Consistent with the radiative drivers from the underlying surface (i.e.,

radiative cooling), during the following December, the atmospheric responses were more zonal during P2 but tended to be

meridional during P1. Moreover, the atmospheric responses during P2 were stronger than those during P1 (Figure 13). The

induced Rossby wave-pattern and anomalous EAJS during P2 (P1) were similar with those in Figures 3 (6) and 4 (7).

Because of the deep anti-cyclonic anomalies over North China and the subtropical West Pacific, the west end of EAJS was

shifted significantly northward, resulting in weak cold air activities during the following December. In the mid-troposphere,

there were Z500 anomaly centers located over West Europe (+), the area north of the Caspian Sea (–), North China and the

Japan Sea (+), and the Aleutian Islands (–). The teleconnected pattern impacted the local meteorological conditions, such as a

shallower boundary layer, small surface wind speed and sufficient water vapor, which confined the ventilation capacities of

the air over the CNC area. The resulting pattern that appeared is shown in Figure 7, and the pattern propagated through the

Mediterranean Sea (+), Northwest Europe (–), the polar region (+), Northeast China and the Okhotsk Sea (–), and the West

Pacific (+) and also existed in Figure 13a. The subtropical high was located over the ocean, and the Aleutian Low extended

westward to the CNC area. The EAJS was enhanced by the significant gradients, indicating obvious meridional cold air

activity. Furthermore, there were no significant responses over the CNC area; thus, the impact on the local ventilation

conditions were not obvious and resulted in a weak relationship with the occurrence of haze.

## 5. Conclusions and discussions

The haze pollution in December has become increasingly serious in the past decade (Figure 1a), and the $DHD_{CNC}$ reached 21

days in 2016. Considering the evident damage of the increasing haze, it is meaningful to study the climatic factors that are





closely related to haze in China. Yin and Wang (2017b) illustrated that the snow cover over East Europe and West Siberia influenced the DHD$_{CNC}$ from 1979 to 2016, but they did not give adequate attention to the physical mechanisms. In this study, we found that the relationship between the SC$_{ES}$ and DHD$_{CNC}$ also varied and was strengthened after the mid-1990s. During 1998–2016, the interannual variation in the SC$_{ES}$ was more significant, and the snow cover with larger interannual variation was distributed zonally and occupied the entirety of East Europe and West Siberia; thus, the forcing effects were more

effective than those during 1979–1997. The associated soil moisture (partially indicating the water effect) and radiation (related with high albedo) were significantly different during these two periods. The radiative cooling effects of the SC$_{ES}$ during the later period were significant and overlapped with the whole target area of snow cover, which was more zonal, broader, and stronger than those in the period of 1979–1997. The soil moisture was also significantly correlated with the SC$_{ES}$, which could last to December between 1998 and 2016. In contrast, there was no close relationship between the

Eurasian soil moisture and the SC$_{ES}$ from 1979 to 1997. Thus, during 1998–2016, the combined influences of the enhanced soil moisture and the radiative cooling that resulted from the positive SC$_{ES}$ anomalies could cause EAJS to shift northward and stimulate the anti-cyclonic anomalies over the CNC area. Under such atmospheric circulations, the local boundary layer was shallower, the surface wind speed was smaller, and the surface moisture was greater. As a result, the atmospheric particles accumulated easily, and the haze occurred frequently. During 1979–1997, both the linkage between the SC$_{ES}$ and

soil moisture and the impacts of soil moisture on atmospheric circulations were negligible. The radiative cooling was the way in which the SC$_{ES}$ modulated the atmospheric circulations. Nevertheless, the correlated regions of radiation were smaller and meridional, and the resulting atmospheric circulations were not significantly linked to the ventilation conditions. Consequently, the relationship between the SC$_{ES}$ and DHD$_{CNC}$ was insignificant from 1979 to 1997 but was strengthened and became significant after the mid-1990s.

In this study, the varied relationship between the SC$_{ES}$ and DHD$_{CNC}$ and the associated physical mechanisms were analyzed, but more detailed investigations, such as the internal processes driving how the soil moisture (radiative cooling) impacted the atmosphere in the following December, should be conducted with numerical models in future work. Many climatic factors at mid-high latitudes have been documented as effective external drivers, including Arctic sea ice (Wang et al 2015, 2016), Eurasian SAT (Yin et al 2017a), SST in the Atlantic (Xiao et al 2015) and Pacific (Yin and Wang 2017a) and Eurasian snow

cover (Yin and Wang 2017b). Some questions raised here include why so many linkages are found in the mid-high latitudes and how they work together to impact the haze pollution in China. This is still an open question that needs to be answered.

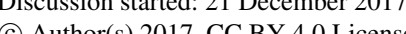

Another question that deserves more attention is why the relationship shifted in approximately 1998. One of the reasonable speculations was the impact of the Pacific Decadal Oscillation, whose phase also shifted in approximately 1998. The questions mentioned above will be addressed in our future work. As a result of the recent enhancement, the significant

relationship possibly improved the potential to predict haze pollution, which is valuable for scientific decision-making related to controlling haze pollution in China.

**Acknowledgements**

This research was supported by the National Natural Science Foundation of China (41705058), the KLME Open Foundation (KLME1607), the CAS–PKU Partnership Program, and the Startup Foundation for Introducing Talent of Nanjing University of Information Science and Technology (20172007)

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

**Figures and Tables caption**

**Table 1** The CC between the $DHD_{CNC}$ and SoilM index in October-November (ON) and December (Dec). OS means 'original sequence', and 'DT' means that the linear trend was removed. '*' indicates the result passed the 95% confidence level, and '**' indicates the CC passed the 99% confidence level.

**Table 2** The CC between the $DHD_{CNC}$ and $I_{LS1}$ ($I_{LS2}$). OS means 'original sequence', and 'DT' means that the linear trend
was removed. '*' indicates the result passed the 95% confidence level, and '**' indicates the CC passed the 99% confidence level.

**Figure 1** (a) the variation of the normalized $DHD_{CNC}$ (black) and $SC_{ES}$ (blue) from 1979 to 2016 after detrending and the 21-yr running correlation coefficient (CC) between the $DHD_{NH}$ and $SC_{ES}$ before (solid, red) and after (dash, red) detrending. (b) The CC between the $DHD_{CNC}$ and snow cover from 1979 to 2016 after detrending. The black dots indicate CCs
exceeding the 95% confidence level (t test). The black box represents the ES area.

Figure 2 The CC between the $SC_{ES}$ and snow cover (a) from 1979 to 1997 and (b) from 1998 to 2016. The black dots indicate the CC exceeded the 95% confidence level (t test). The black box represents the ES area. The linear trend is removed. The green lines indicate that the interannual variations in snow cover were obvious in this region.

**Figure 3** The CC between the $SC_{ES}$ and Z200 (shading) and U200 (contour) in December from 1998 to 2016. The black dots
indicate the CC exceeded the 95% confidence level (t test). The green box represents the ES area. The linear trend is



removed.

**Figure 4** The CC between the $SC_{ES}$ and Z500 (shading, exceeding 90%, 95% and 99% confidence level), stream function (contour), and wave activity flux (arrow) in December from 1998 to 2016. The green box represents the ES area. The linear trend is removed.

**Figure 5** The CC between the $SC_{ES}$ and (a) BLH (shading), surface omega (contour), (b) wind at 850 hPa (arrow), surface wind speed (shading), and surface relative humidity (contour), and (c) SLP (contour) and SAT (shading) in December from 1998 to 2016. The black dots indicate the CC exceeded the 95% confidence level (t test). The linear trend is removed.

**Figure 6** The CC between the $SC_{ES}$ and Z200 (shading) and U200 (contour) in December from 1979 to 1997. The black dots indicate the CC exceeded the 95% confidence level (t test). The green box represents the ES area. The linear trend is removed.

**Figure 7** The CC between the $SC_{ES1}$ and Z500 (shading, exceeding 90%, 95% and 99% confidence level), stream function (contour), and wave activity flux (arrow) in December from 1979 to 1997. The green box represents the ES area. The linear trend is removed.

**Figure 8** The CC between the $SC_{ES1}$ and (a) BLH (shading) and surface omega (contour), (b) wind at 850 hPa (arrow), surface wind speed (shading), and surface relative humidity (contour), and (c) SLP (contour) and SAT (shading) in December from 1979 to 1997. The black dots indicate the CC exceeded the 95% confidence level (t test). The linear trend is removed.

**Figure 9** The CC between the $SC_{ES}$ and soil moisture in (a) October-November and (c) December from 1979 to 1997, and in (b) October-November and (d) December from 1998 to 2016. The black dots indicate the CC exceeded the 95% confidence level (t test). The linear trend is removed. The green boxes are the significantly correlated areas, which were used to calculate the SoilM index.

**Figure 10** The CC between the SoilM index and Z500 (shading, exceeding 90%, 95% and 99% confidence level), stream function (contour), and wave activity flux (arrow) in (a) October-November and (c) December from 1979 to 1997 and in (b) October-November and (d) December from 1998 to 2016. The green box represents the ES area. The linear trend is removed.

**Figure 11** The CC between the $SC_{ES}$ and (a) longwave radiation and (c) shortwave radiation in October-November from





1979 to 1997 and the CC between the SC$_{ES}$ and (b) longwave radiation and (c) shortwave radiation in October-November from 1998 to 2016. The black dots indicate the CC exceeded the 95% confidence level (t test). The linear trend is removed. The green boxes are the significantly correlated areas, which were used to calculate the I$_{LS1}$ (I$_{LS2}$).

**Figure 12** The CC between the SC$_{ES}$ and SAT (shading) and SLP (contour) in October-November (a) from 1979 to 1997 and (b) from 1998 to 2016. The black dots indicate the CC exceeded the 95% confidence level (t test). The green box represents the ES area. The linear trend is removed.

**Figure 13** The CC between (a) I$_{LS1}$, (b) I$_{LS2}$ and Z500 (shading) and U200 (contour) in December. The black dots indicate the CC exceeded the 95% confidence level (t test). The linear trend is removed.

**Figure S1.** The CC between the SC$_{ES}$ and the September (a), October (b) and November (c) Arctic sea ice from 1979 to 1997 after detrending. The black dots indicate the CC exceeded the 95% confidence level (t test).

**Figure S2.** The CC between the SC$_{ES}$ and the September (a), October (b) and November (c) Arctic sea ice from 1998 to 2016 after detrending. The black dots indicate the CC exceeded the 95% confidence level (t test).






**Table 1** The CC between the $DHD_{CNC}$ and SoilM index in October-November (ON) and December (Dec). OS means 'original sequence', and 'DT' means that the linear trend was removed. '*' indicates the result passed the 95% confidence level, and '**' indicates the CC passed the 99% confidence level.

| CC | 1979–1997 | | 1998–2016 | |
|---|---|---|---|---|
| | OS | DT | OS | DT |
| **ON** | 0.15 | 0.14 | 0.77** | 0.69** |
| **Dec** | 0.13 | −0.10 | 0.78** | 0.69** |



**Table 2** The CC between the $DHD_{CNC}$ and $I_{LS1}$ ($I_{LS2}$). OS means 'original sequence', and 'DT' means that the linear trend was removed. '*' indicates the result passed the 95% confidence level, and '**' indicates the CC passed the 99% confidence level.

| CC | 1979–1997 | | 1998–2016 | |
|---|---|---|---|---|
| | OS | DT | OS | DT |
| $I_{LS1}$ | −0.67** | 0.07 | | |
| $I_{LS2}$ | | | −0.78** | −0.72** |






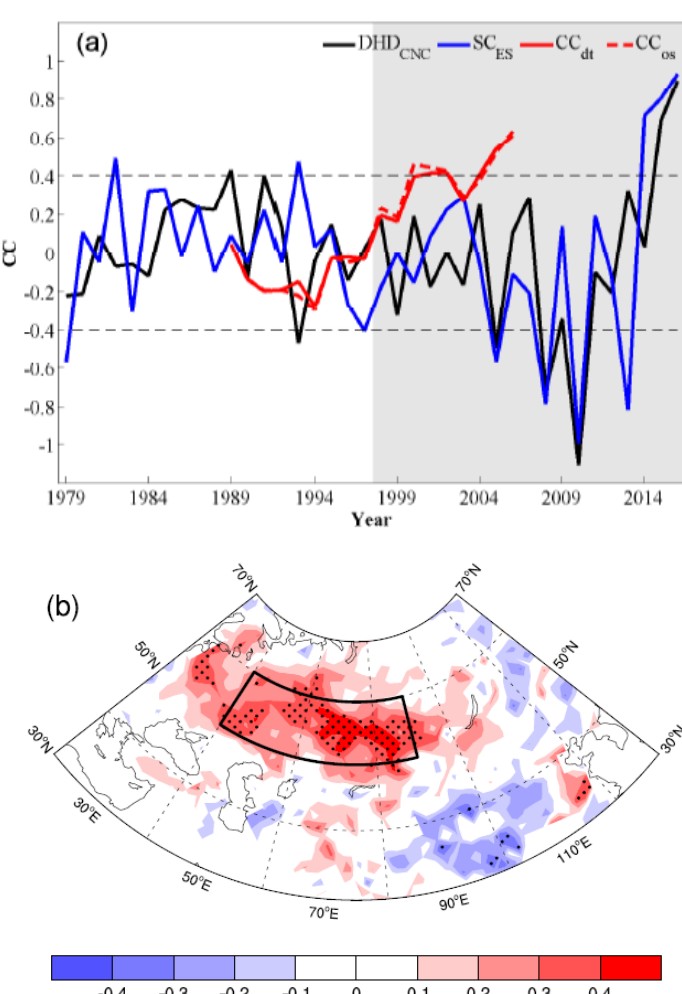


**Figure 1** (a) the variation of the normalized DHD$_{CNC}$ (black) and SC$_{ES}$ (blue) from 1979 to 2016 after detrending and the 21-yr running correlation coefficient (CC) between the DHD$_{NH}$ and SC$_{ES}$ before (solid, red) and after (dash, red) detrending. (b) The CC between the DHD$_{CNC}$ and snow cover from 1979 to 2016 after detrending. The black dots indicate CCs exceeding the 95% confidence level (t test). The black box represents the ES area.







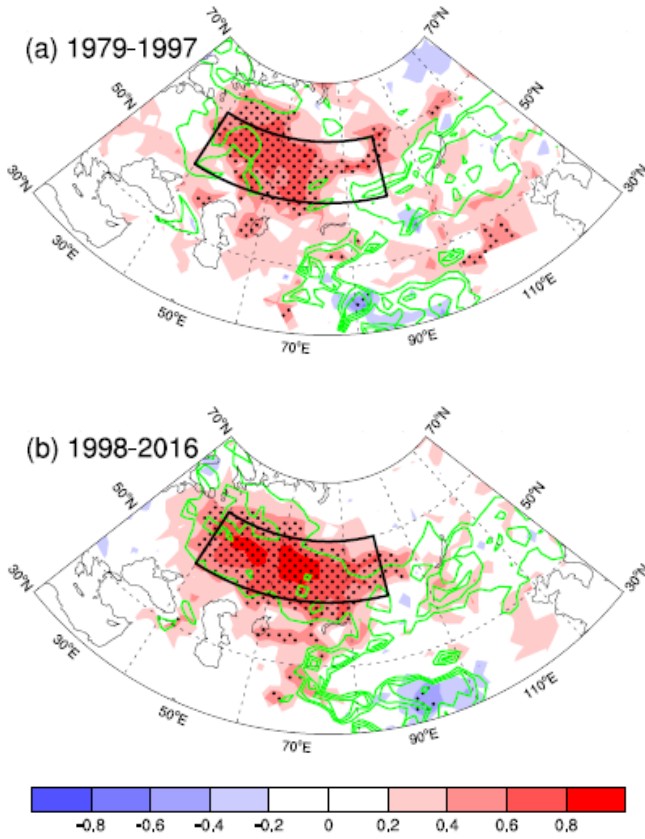

**Figure 2** The CC between the $SC_{ES}$ and snow cover (a) from 1979 to 1997 and (b) from 1998 to 2016. The black dots indicate the CC exceeded the 95% confidence level (t test). The black box represents the ES area. The linear trend is
removed. The green lines indicate that the interannual variations in snow cover were obvious in this region.





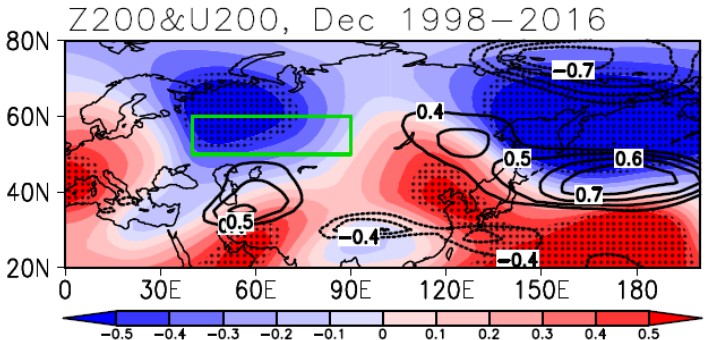


**Figure 3** The CC between the SC$_{ES}$ and Z200 (shading) and U200 (contour) in December from 1998 to 2016. The black dots indicate the CC exceeded the 95% confidence level (t test). The green box represents the ES area. The linear trend is removed.


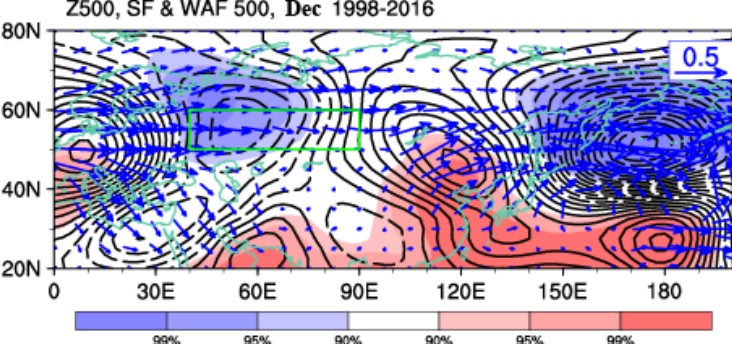

**Figure 4** The CC between the SC$_{ES}$ and Z500 (shading, exceeding 90%, 95% and 99% confidence level), stream function (contour), and wave activity flux (arrow) in December from 1998 to 2016. The green box represents the ES area. The linear 460 trend is removed.






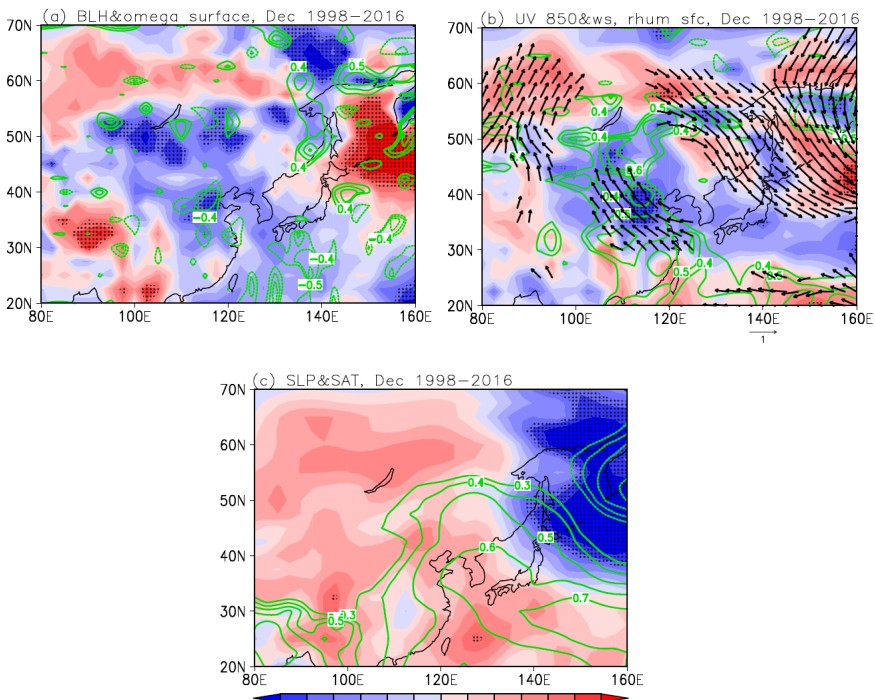

**Figure 5** The CC between the $SC_{ES}$ and (a) BLH (shading), surface omega (contour), (b) wind at 850 hPa (arrow), surface wind speed (shading), and surface relative humidity (contour), and (c) SLP (contour) and SAT (shading) in December from 1998 to 2016. The black dots indicate the CC exceeded the 95% confidence level (t test). The linear trend is removed.







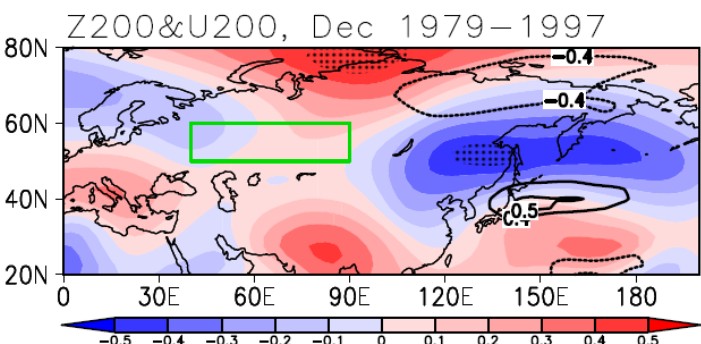

**Figure 6** The CC between the $SC_{ES}$ and Z200 (shading) and U200 (contour) in December from 1979 to 1997. The black dots indicate the CC exceeded the 95% confidence level (t test). The green box represents the ES area. The linear trend is removed.

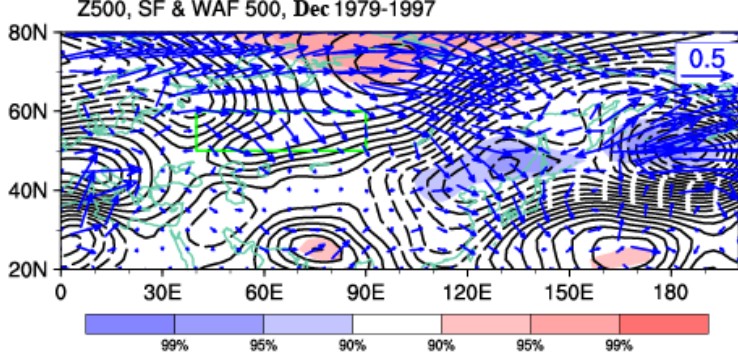


**Figure 7** The CC between the $SC_{ESI}$ and Z500 (shading, exceeding 90%, 95% and 99% confidence level), stream function (contour), and wave activity flux (arrow) in December from 1979 to 1997. The green box represents the ES area. The linear trend is removed.







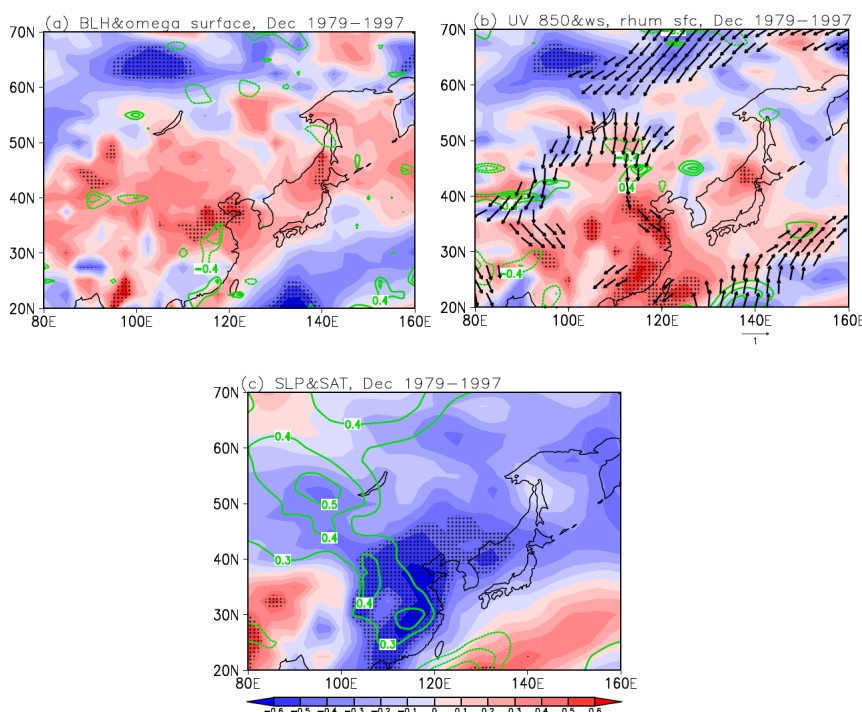

**Figure 8** The CC between the $SC_{ES1}$ and (a) BLH (shading) and surface omega (contour), (b) wind at 850 hPa (arrow), surface wind speed (shading), and surface relative humidity (contour), and (c) SLP (contour) and SAT (shading) in December from 1979 to 1997. The black dots indicate the CC exceeded the 95% confidence level (t test). The linear trend is removed.





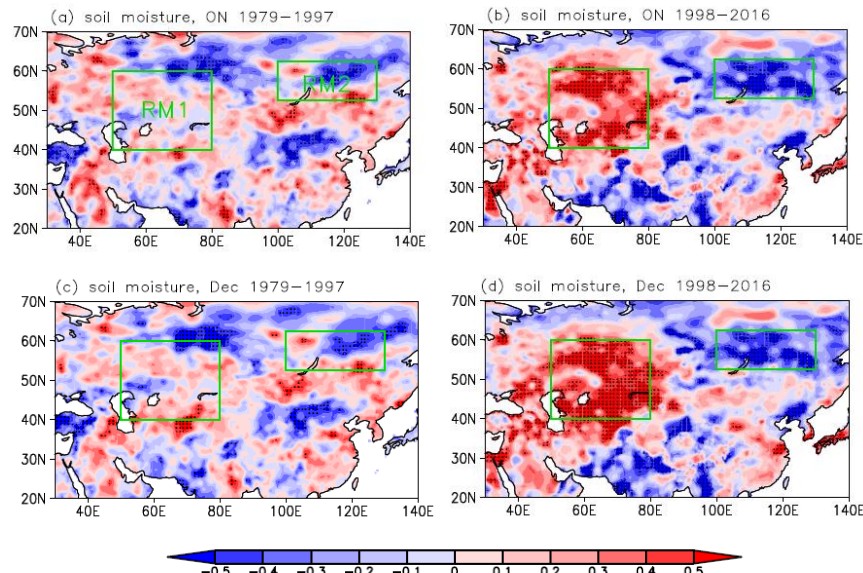


**Figure 9** The CC between the SC$_{ES}$ and soil moisture in (a) October-November and (c) December from 1979 to 1997, and in (b) October-November and (d) December from 1998 to 2016. The black dots indicate the CC exceeded the 95% confidence level (t test). The linear trend is removed. The green boxes are the significantly correlated areas, which were used to

calculate the SoilM index.

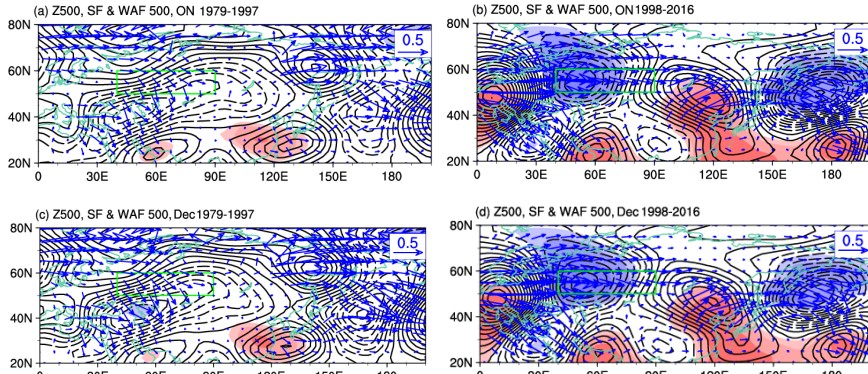

**Figure 10** The CC between the SoilM index and Z500 (shading, exceeding 90%, 95% and 99% confidence level), stream
function (contour), and wave activity flux (arrow) in (a) October-November and (c) December from 1979 to 1997 and in (b) October-November and (d) December from 1998 to 2016. The green box represents the ES area. The linear trend is removed.



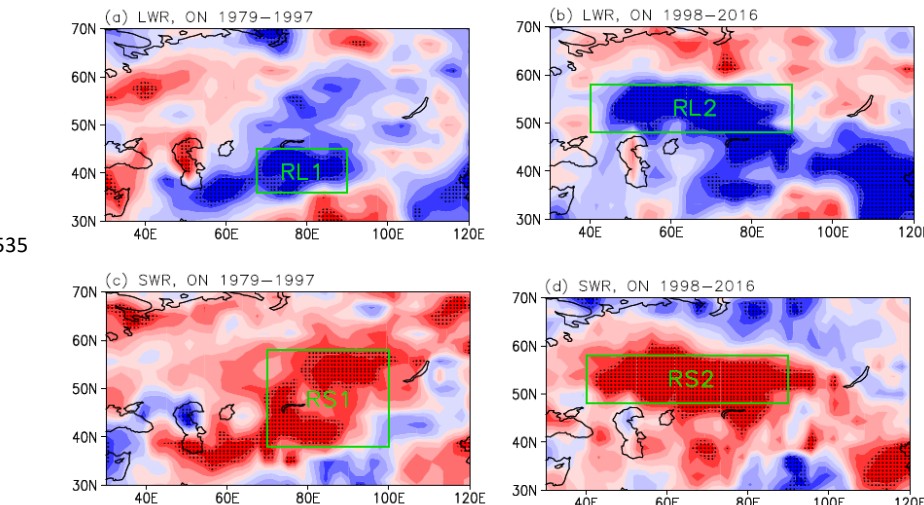


**Figure 11** The CC between the $SC_{ES}$ and (a) longwave radiation and (c) shortwave radiation in October-November from 1979 to 1997 and the CC between the $SC_{ES}$ and (b) longwave radiation and (c) shortwave radiation in October-November from 1998 to 2016. The black dots indicate the CC exceeded the 95% confidence level (t test). The linear trend is removed. The green boxes are the significantly correlated areas, which were used to calculate the $I_{LS1}$ ($I_{LS2}$).









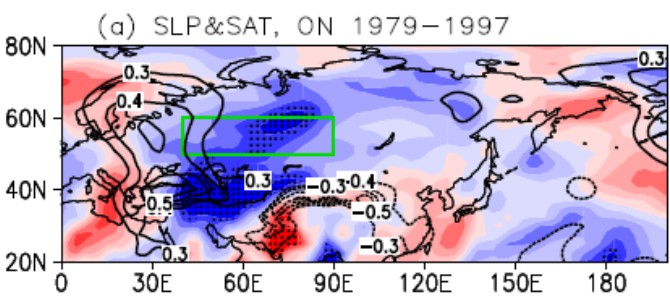

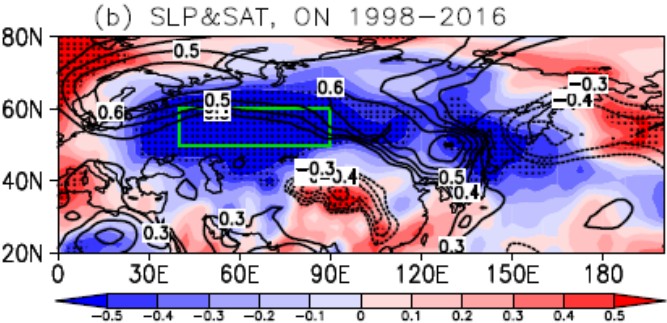

**Figure 12** The CC between the $SC_{ES}$ and SAT (shading) and SLP (contour) in October-November (a) from 1979 to 1997 and
(b) from 1998 to 2016. The black dots indicate the CC exceeded the 95% confidence level (t test). The green box represents
the ES area. The linear trend is removed.







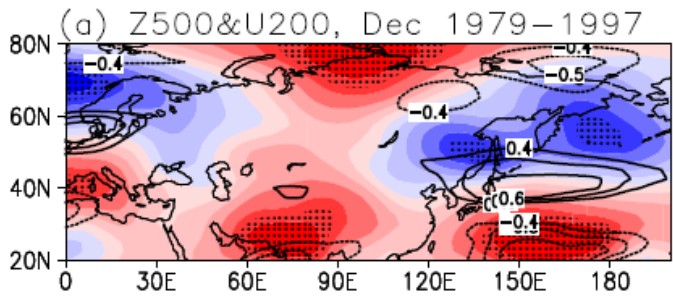

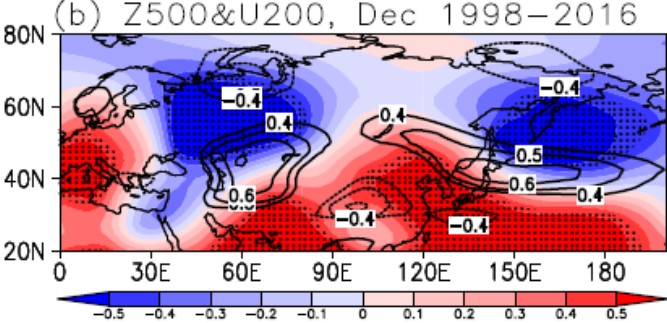

**Figure 13** The CC between (a) $I_{LS1}$, (b) $I_{LS2}$ and Z500 (shading) and U200 (contour) in December. The black dots indicate the CC exceeded the 95% confidence level (t test). The linear trend is removed.
