# Peer review of "The Strengthening Relationship between Eurasian Snow Cover and December Haze Days in Central North China after the Mid-1990s"

_Atmospheric Chemistry and Physics, 2017_

## Referee Comment (RC1) · Anonymous Referee #1 · 5 Jan 2018

In this study, the authors examined relationship between the snow cover over East Europe and West Siberia (SCES) and the number of haze days in December in central North China (DHDCNC). They found changes in SCES can contribute to DHDCNC through influencing soil moisture and land surface radiation during 1998–2016 but the effects are negligible during 1979–1997. This work is interesting and merits publication after following comments addressed.

General Comments:

The authors explained how changes in soil moisture and radiation lead to the atmospheric circulations worsening dispersion conditions. There are a lot of meteorological

fields and effects including in the mechanism. It is better to add a diagram illustrating all the effects.

The authors examined the relationships of SCES and DHDCNC based on the analysis of correlation coefficient. So one question raises, is it possible that they are independent but covary with each other driven by other factors (e.g., climate change).

The author mentioned Eurasian snow cover has been increasing over the last two decades (Cohen et al. 2012). What is the mechanism for the increasing SCES. In addition, based on the positive correlation coefficient between SCES and DHDCNC, does that mean the increasing snow cover in Eurasia lead to the increasing aerosol pollution in Chine during recent decades? If so, it is more interesting and the authors may discuss more about it.

Specific comments:

Line 26: typo '/'.

Line 29: Recent studies found, in additions to emissions and climate change, aerosol-meteorology feedbacks have contributed the haze in China (e.g., Ding et al., 2016; Yang et al., 2017a).

Line 37: Besides to less ventilation, transport of aerosols from upwind can also lead to regional aerosol pollution (e.g., Yang et al., 2017b).

Line 124: How can surface upward motion appears in a sinking motion atmosphere. Upward motion sometimes means non-stagnation and strong dispersion. How can it accumulate aerosols?

Line 166: Why the "west wet-east dry" pattern leads to poor dispersion conditions?

Line 172: What is the direction for positive longwave and shortwave radiation defined in this study? Also, the authors should make clear that they are the surface net radiative fluxes.

References

Ding, A. J., X. Huang, W. Nie, J. N. Sun, V.-M. Kerminen, T. Petäjä, H. Su, Y. F. Cheng, X.-Q. Yang, M. H. Wang, et al. (2016), Enhanced haze pollution by black carbon in megacities in China, Geophys. Res. Lett., 43, 2873–2879, doi: 10.1002/2016GL067745.

Yang, Y., L. M. Russell, S. Lou, H. Liao, J. Guo, Y. Liu, B. Singh, and S. J. Ghan, Dust-wind interactions can intensify aerosol pollution over eastern China, Nat. Commun., 8, 15333, doi:10.1038/ncomms15333, 2017a.

Yang, Y., Wang, H., Smith, S. J., Ma, P.-L., and Rasch, P. J.: Source attribution of black carbon and its direct radiative forcing in China, Atmos. Chem. Phys., 17, 4319-4336, https://doi.org/10.5194/acp-17-4319-2017, 2017b.

---

## Referee Comment (RC2) · Anonymous Referee #2 · 26 Jan 2018

This study discussed the effect of Eurasian snow cover on December haze days. Recently, severe haze occurs in the broad area of China, and the discussion of the relationship between Eurasian snow cover on December haze days is helpful to understand the mechanism modulation the formation of haze. The topic is interesting and I have a few questions listed below: 1. Line 124 The authors said "In contrast, the associated vertical velocity at the surface was upward, indicating an ascending motion near the surface." I think the downward vertical velocity favors the haze formation due to weak dispersion conditions. The authors published a paper in 2017 (Atmos. Chem. Phys., 17, 11673–11681, 2017 https://doi.org/10.5194/acp-17-11673-2017), and in Figure 7, the omega was positive, and the authors stated that "Under their influence, there was

a descending motion from 30 to 55_ N (Fig. 7)," and claimed this condition support the severe haze. Thus, the statement regarding the vertical motion in this study is somewhat contradicts with the previous study.

2. References: It is easier to read if a few spaces were left in front of the first line of each reference. Alternatively, a number can be used to separate each reference as well.

3. Line 59: "Basing on" should be "based on"

4. Line 100-103: The authors declared that "during P2, the snow cover with larger interannual variation was distributed widely and zonally": do you have a figure displaying the distributions of the snow cover? It is hard to tell without a figure how the snow cover was spatially distributed.

5. There are a few places which did not clearly mention the figure number, which makes it hard to follow. For example: Line 145: In the first paragraph of the section 4 "possible physical mechanisms", the authors should mention Figure 9 first, so the readers can follow the authors easily. Otherwise, it is hard to know which figure the authors are referring to. Line 176: RL1. . ., this information is from Figure 11a, so the authors should point out Fig. 11a immediately after the description.

6. Line 180: if there was more SCES, the absolute value of the net longwave radiation and net shortwave radiation would both be smaller. The signs of the correlations between SCES and net longwave radiation, SCES and net shortwave radiation are opposite. I am not sure why the absolute value of the net longwave radiation and net shortwave radiation would both be smaller when there was more SCES

7. Line 197: EAJS was shifted significantly northward Without a base location, how can this shift be identified?

8. The figure qualities and descriptions of captions need to be improved: For example: Figure 1: CCdt, CCOS should be explained in the caption. A figure is in principle can

be independent from the paper. Thus, one should get all the information from the figure or caption without searching from the main text.

---

## Author Comment (AC1) · 27 Feb 2018

In this study, the authors examined relationship between the snow cover over East Europe and West Siberia (SCES) and the number of haze days in December in central North China (DHDCNC). They found changes in SCES can contribute to DHDCNC through influencing soil moisture and land surface radiation during 1998–2016 but the effects are negligible during 1979–1997. This work is interesting and merits publication after following comments addressed.

General Comments:

1. **The authors explained how changes in soil moisture and radiation lead to the atmospheric circulations worsening dispersion conditions. There are a lot of meteorological fields and effects including in the mechanism. It is better to add a diagram illustrating all the effects.**

*Reply:*

**A new diagram was drawn to make the understanding easier**. In "Conclusions and Discussions", we mentioned that "To exemplify the associated mechanisms during 1998–2016, a diagram was drawn and supplemented as Figure S3."

*Revisions:*

In "Conclusions and Discussions"

……To exemplify the associated mechanisms during 1998–2016, a diagram was drawn and supplemented as Figure S3.

[Figure]

**Figure S3.** Diagram of the associated physical mechanisms. Near surface, the ON radiation (contour) and soil moisture (shade) were influenced by the $SC_{ES}$. On the mid-high level, the teleconnected Rossby wave-like pattern propagated into the Central North China, representing by Z500 (shade), stream function (contour) and wave activity flux (arrow). Finally, the local anti-cyclonic circulation near surface (arrow) led to weak ventilation conditions in December.

2. **The authors examined the relationships of SCES and DHDCNC based on the analysis of correlation coefficient. So one question raises, is it possible that they are independent but covary with each other driven by other factors (e.g., climate change).**

*Reply:*

The emphasis of this study was the **interannual variation** of haze days and its relationship with snow cover. Thus, during analyzing, the linear trend was removed.

In our study, we assumed that the energy consumption linearly increased in the recent years. On such hypothesis, the human activities (also including **the climate change**) mainly impacted the long-term trend of $DHD_{CNC}$. **After removal of the linear trend, the interannual variability of haze pollution should be mainly the result of climatic anomalies.**

In the revisions, the authors also **supplemented some previous studies on the long-term trend of haze pollutions**, i.e., the impacts of climate change.

*Revisions:*

In "Introduction"

……For the long-term trend of haze pollution, Wang and Chen (2016) illustrated the roles of climate change on the in eastern China and emphasized the effects of the Arctic sea ice. Cai et al (2017) analyzed the weather conditions conductive to Beijing severe haze more frequent under climate change. There were also previous studies on the interannual variation of haze and associated climatic conditions. The possible physical processes in the atmosphere that caused this the haze events……

3. **The author mentioned Eurasian snow cover has been increasing over the last two decades (Cohen et al. 2012). What is the mechanism for the increasing SCES.**

*Reply:*

**An associated reference** was cited to answer this question. "Different from the declining trend of Arctic sea ice, Eurasian snow cover has been increasing over the last two decades (Cohen et al. 2012), probably due to **the increased southward**

**moisture transport from the melted Arctic Ocean** (Deser et al., 2010)."

*Revisions:*

In "Introduction"

Different from the declining trend of Arctic sea ice, Eurasian snow cover has been increasing over the last two decades (Cohen et al. 2012), probably due to the increased southward moisture transport from the melted Arctic Ocean (Deser et al., 2010)……

Deser, C., R. Tomas, M. Alexander, and D. Lawrence. 2010. The seasonal atmospheric response to projected Arctic sea ice loss in the late twentyfirst century, J. Clim., 23(2), 333–351, doi:10.1175/2009JCLI3053.1.

4. **In addition, based on the positive correlation coefficient between SCES and DHDCNC, does that mean the increasing snow cover in Eurasia lead to the increasing aerosol pollution in China during recent decades? If so, it is more interesting and the authors may discuss more about it.**

*Reply:*

In this manuscript, we used the haze days to **represent the general of haze pollution**. During 1998–2016, the accumulated snow cover significantly intensified the haze pollution in Central North China by atmospheric teleconnection.

Specific comments:

**Line 26: typo '/'.**

*Reply:*

The error has been corrected.

*Revisions:*

In December 2016, central North China (CNC, located at 30–41 °N, 110–120 °E),

**Line 29: Recent studies found, in additions to emissions and climate change, aerosolmeteorology feedbacks have contributed the haze in China (e.g., Ding et al., 2016;Yang et al., 2017a).**

*Reply:*

The advice was adopted. Some revisions and new references were added.

*Revisions:*

In "Introduction"

Beyond anthropogenic emissions, the atmospheric circulations (Yin and Wang 2017b) and **aerosol-meteorology feedback (Ding et al. 2016, Yang et al. 2017a)** have significantly contributed the severe haze in China.

Yang, Y., L. M. Russell, S. Lou, H. Liao, J. Guo, Y. Liu, B. Singh, and S. J. Ghan, Dustwind interactions can intensify aerosol pollution over eastern China, Nat. Commun., 8, 15333, doi:10.1038/ncomms15333, 2017a.

**Line 37: Besides to less ventilation, transport of aerosols from upwind can also lead to regional aerosol pollution (e.g., Yang et al., 2017b).**

*Reply:*

The advice was adopted. Some revisions and new references were added.

*Revisions:*

In "Introduction"

……Furthermore, the southerly anomalies that are characteristic of East Asian winter monsoons (Li et al. 2015; Yin et al. 2015) may have weakened the cold air and wind speed but enhanced **the transportation of humid air flow and aerosols (Yang et al. 2017b)**……

Yang, Y., Wang, H., Smith, S. J., Ma, P.-L., and Rasch, P. J.: Source attribution of black carbon and its direct radiative forcing in China, Atmos. Chem. Phys., 17, 4319-4336, https://doi.org/10.5194/acp-17-4319-2017, 2017b.

**Line 124: How can surface upward motion appears in a sinking motion atmosphere. Upward motion sometimes means non-stagnation and strong dispersion. How can it accumulate aerosols?**

*Reply:*

We have corrected the discussions about the vertical motion.

1. There was significant **upward motion near surface (Figure 5a), indicating weak convergences of the aerosols discharged in the circumjacent regions**. Actually, in winter, the weak convergence near surface was a classical synoptic situation resulting in severe haze pollution. This convergence could transport the aerosols discharged in the surrounding to the CNC area, but cannot disturbed the shallow boundary layer. **The converging and local aerosols both accumulated and reached a high concentration**.

2. The description of the sinking motion on the **mid-high level** was not precise and has been **deleted** from this manuscript on the premise that the conclusions were not affected. In a recent study, we have found that **the vertical motions below different parts of the anti-cyclonic circulation were also different**. It is inaccurate to simply describe the associated vertical velocity as sinking or ascending motion. Thus, we are going to write a new manuscript to discuss the associated vertical motions.

*Revisions:*

In "Possible physical mechanisms"

……The associated vertical velocity at the surface was upward (Figure 5a), indicating weak convergences of the aerosols discharged in the circumjacent regions. However, due to the shallower planetary boundary layer (Figure 5a), the converging and local aerosols cannot be dispersed into the upper atmosphere. The local convergences, combined with the weak surface wind (Figure 5b), easily enabled aerosols to accumulate over the CNC area……

 lead to the shallower planetary boundary layer (Figure 5a) The associated vertical velocity at the surface was upward (Figure 5a), indicating  convergences of the aerosols discharge in the circumjacent regions. However,  due to the shallower planetary boundary layer (Figure 5a), the converging and local aerosols cannot be dispersed into the upper atmosphere. The local convergences, combined with the weak  surface wind (Figure 5b), easily enabled aerosols to accumulate over the CNC area. Near the surface, the positive SLP anomalies were situated in the east of China and the western Pacific (Figure 5c). The stimulated southerlies overlapped with

**Line 166: Why the "west wet-east dry" pattern leads to poor dispersion conditions?**

*Reply:*

This is still an open question and beyond the scope of this study. In the final section, we discussed that this question should be **a future work** with numerical models.

*Revisions:*

In "Conclusions and Discussions"

……In this study, the varied relationship between the $SC_{ES}$ and $DHD_{CNC}$ and the associated physical mechanisms were analyzed, but more detailed investigations, **such as the internal processes driving how the soil moisture (radiative cooling) impacted the atmosphere in the following December, were not included in this study and should be conducted with numerical models in future work**……

**Line 172: What is the direction for positive longwave and shortwave radiation defined in this study? Also, the authors should make clear that they are the surface net radiative fluxes.**

*Reply:*

**The upward radiation is positive**. The description on the radiation dataset was improved to make it clearer.

*Revisions:*

In "Datasets and methods"

……the vertical wind, the **surface** net longwave radiation and the **surface** net shortwave radiation **(upward radiation is positive)** data were downloaded from……

---

## Author Comment (AC2) · 27 Feb 2018

This study discussed the effect of Eurasian snow cover on December haze days. Recently, severe haze occurs in the broad area of China, and the discussion of the relationship between Eurasian snow cover on December haze days is helpful to understand the mechanism modulation the formation of haze. The topic is interesting and I have a few questions listed below:

1.  **Line 124: The authors said "In contrast, the associated vertical velocity at the surface was upward, indicating an ascending motion near the surface." I think the downward vertical velocity favors the haze formation due to weak dispersion conditions. The authors published a paper in 2017 (Atmos. Chem. Phys.,17, 11673–11681, 2017 https://doi.org/10.5194/acp-17-11673-2017), and in Figure 7, the omega was positive, and the authors stated that "Under their influence, there was a descending motion from 30 to 55_ N (Fig. 7)," and claimed this condition support the severe haze. Thus, the statement regarding the vertical motion in this study is somewhat contradicts with the previous study.**

*Reply:*

We have corrected the discussions about the vertical motion.

1. There was significant **upward motion near surface (Figure 5a), indicating weak convergences of the aerosols discharged in the circumjacent regions**. Actually, in winter, the weak convergence near surface was a classical synoptic situation resulting in severe haze pollution. This convergence could transport the aerosols discharged in the surrounding to the CNC area, but cannot disturbed the shallow boundary layer. **The converging and local aerosols both accumulated and reached a high concentration**.

2. The description of the sinking motion on the **mid-high level** was not precise and has been **deleted** from this manuscript on the premise that the conclusions were not affected. In a recent study, we have found that **the vertical motions below different parts of the anti-cyclonic circulation were also different**. It is inaccurate to simply describe the associated vertical velocity as sinking or ascending motion. Thus, we are going to write a new manuscript to discuss the associated vertical motions.

*Revisions:*

In "Possible physical mechanisms"

……The associated vertical velocity at the surface was upward (Figure 5a), indicating weak convergences of the aerosols discharged in the circumjacent regions. However, due to the shallower planetary boundary layer (Figure 5a), the converging and local aerosols cannot be dispersed into the upper atmosphere. The local convergences, combined with the weak surface wind (Figure 5b), easily enabled aerosols to accumulate over the CNC area……

The sinking motion caused by these anti-cyclonic anomalies could _lead to the shallower planetary boundary layer (Figure 5a) and the rather weak dispersion capacity of atmospheric particulates. In contrast, tThe associated vertical velocity at the surface was upward (Figure 5a), indicating an ascending motion _convergences of the aerosols discharge in the circumjacent regions. However, near the surface, due to the shallower planetary boundary layer (Figure 5a), the converging and local aerosols cannot be dispersed into the upper atmosphere. The local rising airconvergences, combined with the weak south surface wind (Figure 5b), easily enabled aerosols to accumulate over the CNC area. Near the surface, the positive SLP anomalies were situated in the east of China and the western Pacific (Figure 5c). The stimulated southerlies overlapped with

2. **References: It is easier to read if a few spaces were left in front of the first line of each reference. Alternatively, a number can be used to separate each reference as well.**

*Reply:*

The advice was adopted.

*Revisions:*

under Climate Change. Nature Climate Change. doi:10.1038/nclimate3249

Chen H. P., Wang H. J. 2015. Haze days in North China and the associated atmospheric circulations based on daily visibility data from 1960 to 2012. J. Geophys. Res. Atmos. 120(12): 5895–5909 DOI: 10.1002/2015JD023225.

Cohen J L, Furtado J C, Barlow M A, Alexeev V A, Cherry JE. 2012. Arctic warming, increasing snow cover and widespread boreal winter cooling. Environ Res Lett 7:014007

Cohen J, Barlow M, Kushner PJ, Saito K. 2007. Stratosphere and troposphere coupling and links with Eurasian land surface variability. J Climate. 20:5335–5343

Dee D. P., Uppala S. M., Simmons A. J., Berrisford P., Poli P., Kobayashi S., Andrae U., Balmaseda M. A., Balsamo G.,

**3. Line 59: "Basing on" should be "based on"**

*Reply:*

The error has been corrected.

*Revisions:*

……Zou et al (2017) also pointed out that there was close relationship between Eurasia snow and haze in China **based on** the observational and numerical analysis……

**4. Line 100-103: The authors declared that "during P2, the snow cover with larger interannual variation was distributed widely and zonally": do you have a figure displaying the distributions of the snow cover? It is hard to tell without a figure how the snow cover was spatially distributed.**

*Reply:*

Due to our poor expression, there was some confusion. What we wanted to show was **the intensity of interannual variations**. In the revision, we clarified the intensity of the interannual variations was described by the **standard deviation in Figure 2 (green lines)**.

*Revisions:*

In "Strengthening relationship and associated atmospheric circulations"

……However, during P1, the CC over the east part of the ES area was insignificant. **The intensity of the interannual variations (i.e., expressed by the standard deviation in Figure 2)** in snow cover over the Tibet Plateau and Mongolian Plateau were evident both during P1 and P2……

[Figure]

[Figure]

**Figure 2** The CC between the $SC_{ES}$ and snow cover (a) from 1979 to 1997 and (b) from 1998 to 2016. The black dots indicate the CC exceeded the 95% confidence level (t test). The black box represents the ES area. The linear trend is removed. **The green lines indicate that the interannual variations in snow cover were obvious in this region.**

5. **There are a few places which did not clearly mention the figure number, which makes it hard to follow. For example: Line 145: In the first paragraph of the section 4 "possible physical mechanisms", the authors should mention Figure 9 first, so the readers can follow the authors easily. Otherwise, it is hard to know which figure the authors are referring to. Line 176: RL1. . ., this information is from Figure 11a, so the authors should point out Fig. 11a immediately after the description.**

*Reply:*

The error has been corrected. Furthermore, the similar errors were checked and revised throughout the manuscript.

*Revisions:*

In "Possible physical mechanisms"

……The associated anomalous circulations tended to lead local meteorological conditions (e.g., higher BLH and more obvious surface wind speed) to favor ventilation (**Figure 8**), which was consistent with the 21-yr running CC in Figure 1a (i.e., negative before the mid-1990s)……

……During P2, the SCES was significantly positively correlated with soil moisture around the Caspian Sea, Balkhash Lake, and Ural Mountains (**Figure 9**, RM1: 50–80oE, 40–60 °N)……

……this was denoted as RS1 (70–100 E, 38–58 N) and was mountainous (**Figure 11c**). In contrast, the regions that had significant and negative CCs and net longwave radiation were smaller and over the Pamir Mountains (**Figure 11a**, RL1: 67.5–90 E, 36–45 N). By contrast, the significant correlated regions with net longwave radiation (**Figure 11b**, RL2) and net shortwave radiation **(Figure 11d**, RS2) were the same and nearly overlapped with the ES area during P2, which was wider and had a zonal distribution……

6.  **Line 180: if there was more SCES, the absolute value of the net longwave radiation and net shortwave radiation would both be smaller. The signs of the correlations between SCES and net longwave radiation, SCES and net shortwave radiation are opposite. I am not sure why the absolute value of the net longwave radiation and net shortwave radiation would both be smaller when there was more SCES**

*Reply:*

**The upward radiation is positive.** Shown by the Figure below, the net surface short wave radiation was globally negative. However, the net surface long wave radiation was globally positive. Thus, the more significant positive correlation with short wave radiation and negative correlation with long wave radiation both meant the radiation reduced, i.e., the net shortwave and net longwave radiations were both reduced.

To make the analysis clearer, the writing was improved both in the section "dataset" and in the section "physical mechanism".

[Figure]

Figure the net surface short wave radiation (left) and net surface long wave radiation (right) in December, directly plotted by the website of NOAA/NCEP PSD

*Revisions:*

In "Datasets and methods"

……the vertical wind, the **surface** net longwave radiation and the **surface** net shortwave radiation **(upward radiation is positive)** data were downloaded from……

In "Possible physical mechanisms"

……As a feedback, the outgoing longwave radiations emitted by the cooler land surface were weakened and had radiative cooling impacts on the atmosphere (Zhang et al. 2017). **That is to say: the absorbed shortwave and outgoing longwave radiations were both reduced**……

……According to the above analysis, if there was more $SC_{ES}$, the net shortwave and net longwave radiations were both reduced, i.e., the absolute value of the net longwave radiation and net shortwave radiation would both be smaller……

7. **Line 197: EAJS was shifted significantly northward Without a base location, how can this shift be identified?**

*Reply:*

The climatic distribution of U200 was showed below. The location of EAJS was around 30ºN. In Figure 3, the U200 anomalies were negative near 30ºN, but positive northward. Thus, EAJS was shifted significantly northward. If, we plotted the climatic distribution of U200 in the Figures of the manuscript, the Figures should become too complicated and take up too much space. After careful thought, the climatic distribution of U200 was still omitted.

[Figure]

Figure the climatic distribution of U200, directly plotted by the website of NOAA/NCEP PSD

[Figure]

Figure 3 The CC between the SCES and Z200 (shading) and U200 (contour) in December from 1998 to 2016. The black dots indicate the CC exceeded the 95% confidence level (t test). The green box represents the ES area. The linear trend is removed.

8. **The figure qualities and descriptions of captions need to be improved: For example: Figure 1: CCdt, CCOS should be explained in the caption. A figure is in principle can be independent from the paper. Thus, one should get all the information from the figure or caption without searching from the main text.**

*Reply:*

The error has been corrected. Furthermore, the similar errors were checked and revised throughout the manuscript.

*Revisions:*

**Figure 1** (a) the variation of the normalized DHD$_{CNC}$ (black) and SC$_{ES}$ (blue) from 1979 to 2016 after detrending and the 21-yr running correlation coefficient (CC) between the DHD$_{NH}$ and SC$_{ES}$ before (solid, red) and after (dash, red) detrending. (b) The CC between the DHD$_{CNC}$ and snow cover from 1979 to 2016 after detrending. The black dots indicate CCs exceeding the 95% confidence level (t test). The black box represents the ES area. **The subscript "dt" and "OS" in panel (a) represented the CC was calculated by the detrending and original sequence.**

**Figure 9** The CC between the SC$_{ES}$ and soil moisture in (a) October-November and (c) December from 1979 to 1997, and in (b) October-November and (d) December from 1998 to 2016. The black dots indicate the CC exceeded the 95% confidence level (t test). The linear trend is removed. The green boxes **(RM1 and RM2)** are the significantly correlated areas, which were used to calculate the SoilM index.

**Figure 11** The CC between the $SC_{ES}$ and (a) longwave radiation and (c) shortwave radiation in October-November from 1979 to 1997 and the CC between the $SC_{ES}$ and (b) longwave radiation and (c) shortwave radiation in October-November from 1998 to 2016. The black dots indicate the CC exceeded the 95% confidence level (t test). The linear trend is removed. The green boxes **(RL and RS)** are the significantly correlated areas, which were used to calculate the $I_{LS1}$ ($I_{LS2}$).